# Studying the Effect of Electrode Material and Magnetic Field on Hydrogen Production Efficiency

**Yen-Ju Chen** [1], **Yan-Hom Li** [1,2,*] and **Ching-Yao Chen** [1]

[1] Department of Mechanical Engineering, National Yang Ming Chiao Tung University, Hsinchu 300, Taiwan; yenju.c@nycu.edu.tw (Y.-J.C.); chingyao@nycu.edu.tw (C.-Y.C.)

[2] Department of Mechanical and Aerospace Engineering, Chung-Cheng Institute of Technology, National Defense University, Taoyuan 335, Taiwan

[*] Correspondence: athomeli@ccit.ndu.edu.tw

**Abstract:** Water electrolysis is one of the most common methods to produce hydrogen gas with high purity, but its application is limited due to its low energy efficiency. It has been proved that an external magnetic field can reduce energy consumption and increase hydrogen production efficiency in water electrolysis. In this study, electrodes with different magnetism were subjected to a perpendicular magnetic field for use in hydrogen production by water electrolysis. Gas bubbles that evolve from the surface of a horizontal electrode detach faster than the bubbles from a vertical electrode. The locomotion of the bubbles is facilitated if the horizontal electrode faces a magnet, which induces the revolution of bubbles between the electrodes. However, the magnetic field does not increase the current density effectively if the electrodes are more than 5 cm apart. A paramagnetic (platinum) electrode has a more significant effect on bubble locomotion than a diamagnetic (graphite) material and is able to increase the efficiency of electrolysis more effectively when a perpendicular magnetic field is applied. The conductivity of platinum electrodes that face a magnet increases if the distance between the electrodes is less than 4 cm, but the conductivity of graphite electrodes does not increase until the inter-electrode distance is reduced to 2 cm. On the other hand, horizontal graphite electrodes that are subjected to a perpendicular magnetic field will generate a higher gas production rate than a platinum electrode without a magnetic field if the inter-electrode distance is less than 1 cm.

**Keywords:** water electrolysis; hydrogen; MHD; bubble revolution; magnetic field

## 1. Introduction

Hydrogen is a clean energy carrier with the highest specific energy density and is the preferred alternative to fossil fuels to satisfy the huge global energy demand because the energy-releasing process is environmentally friendly. Hydrogen gas can be produced using alkaline water electrolysis (AWE) [1–4], proton exchange membrane (PEM) electrolysis [5–8], and a solid oxide electrolyzer cell (SOEC) [9,10]. Water electrolysis is commonly used because it requires simple equipment and produces very pure hydrogen. However, oxygen and hydrogen bubbles that respectively evolve from the anode and cathode will adhere to the surface of electrodes and block the subsequent electrochemical reaction. Thus, the water electrolysis process has low efficiency [11,12], with an overall production rate that is as low as 4% [13]. On the other hand, renewable energy has been extensively explored for the efficiency improvement of green hydrogen production due to the increase in energy demand and environmental impacts [14–16].

Numerous studies have shown that using a magnetic field increases hydrogen production in practical applications [17–19]. The facilitating mechanism of the magnetic field is the Lorentz force, which is also described as a magnetohydrodynamic (MHD) force. The Lorentz force generates convection that enhances mass transfer and reduces ohmic voltage drop and concentration overpotential [20,21]. Ferromagnetic materials such as nickel are

better cathode materials than paramagnetic (platinum) and diamagnetic (graphite) materials under a magnetic field [20]. Furthermore, convective flow also increases the desorption of bubbles and reduces their coverage in electrodes [17,22,23]. It has been demonstrated that varying the magnetic field configuration produces various convective flow patterns that have different effects on water electrolysis [23].

The magnitude of the magnetic field affects hydrogen production in the water electrolysis process, but the most influential factor is the gradient magnetic field that is used for the electrolysis of water [24]. A magnetic field that is normal to the surface of the horizontal electrode causes bubbles to revolve and spread between the electrodes, which increases the efficiency of electrolysis [23]. The effect of a gradient magnetic field on electrochemical processes has been studied [25–27] in terms of the effect of various gradients for the magnetic flux density or the field intensity. However, studies have rarely reported the effect of magnetism or the layout of the electrode and the current density.

Nickel shows a high initial electrocatalytic activity towards the hydrogen evolution reaction (HER); however, it experiences extensive deactivation as a cathode during water electrolysis [28]. It is well known that platinum (Pt) is the best electrocatalyst currently available. However, its high cost and scarcity have hindered the commercialization of many green technologies that require the use of Pt electrocatalysis [29]. On the other hand, the graphite contained in activated carbon has magnetic potential since it has aromatic rings [30]. This study used platinum and graphite electrodes in different positions for hydrogen production using water electrolysis. The movement of the bubbles under the effect of electrodes with different magnetism and inter-electrode distances is demonstrated. The essential controlling parameters for hydrogen production are also proposed to determine the optimal electrode layout for hydrogen production under a magnetic field.

## 2. Experimental Setup and Methods

The experimental setup for this study is similar to the apparatus in [23]. Electrode materials with different magnetism were used to determine the effect of paramagnetism and diamagnetism on the hydrogen evolution reaction. The paramagnetic and diamagnetic electrodes were respectively made of platinum and graphite materials with an area of 50 mm × 50 mm (Guang Yi, Changhua, Taiwan) and immersed in distilled water without mixing any acid or alkali solution at room temperature. The experiments were conducted in a water tank with dimensions of 400 mm × 240 mm × 200 mm. Two pieces of N35 NdFeB magnet with a surface magnetic field strength of 0.22 T and dimensions of 100 mm × 50 mm × 10 mm were used to generate a magnetic field perpendicular to the electrodes. A direct current (DC) power supply (GWInstek APS-1102, GWinstek, New Taipei, Taiwan) was used to provide a fixed voltage, and the current density was measured under various experimental conditions.

Water electrolysis is the process whereby water is split into hydrogen and oxygen with the application of electrical energy. For distilled water electrolysis, the acid-balanced half-reactions for the HER at the cathode and the oxygen evolution reaction (OER) at the anode are written as:

$$2H^+ + 2e^- \rightarrow H_2 \tag{1}$$

$$2H_2O \rightarrow O_2 + 4H^+ + 4e^- \tag{2}$$

The global reaction for the two cases is given by:

$$2H_2O \rightarrow 2H_2 + O_2 \tag{3}$$

During water electrolysis, gas production is proportional to the electric current. This study plots the I-V curves and compares the conductivity of the electrolyte for different electrode materials, electrode layouts, and inter-electrode distances with/without a magnetic field.

### 3. Results and Discussion

*3.1. Influence of Electrode Layouts on the Detachment of Gas Bubbles*

The buoyancy force significantly affects the detachment of bubbles that evolve from the electrode, so the direction of the electrode placement has a significant effect on the efficiency of water electrolysis. For a vertically placed electrode, as shown in Figure 1a, gas bubbles do not evolve from either the anode or the cathode in significant amounts at the beginning. Oxygen and hydrogen bubbles respectively emerge from the anode (left) and cathode (right) surface at t = 3 s. More hydrogen bubbles are produced than oxygen bubbles, which is consistent with the global reaction for water electrolysis, as shown in (3). Some bubbles detach from the electrode surface and move upward and accumulate at the two upper corners because of the buoyancy force. However, numerous bubbles adhere to the electrode surface, particularly the cathode, so the resistance between the electrodes increases, and a higher charging voltage is required to produce hydrogen.

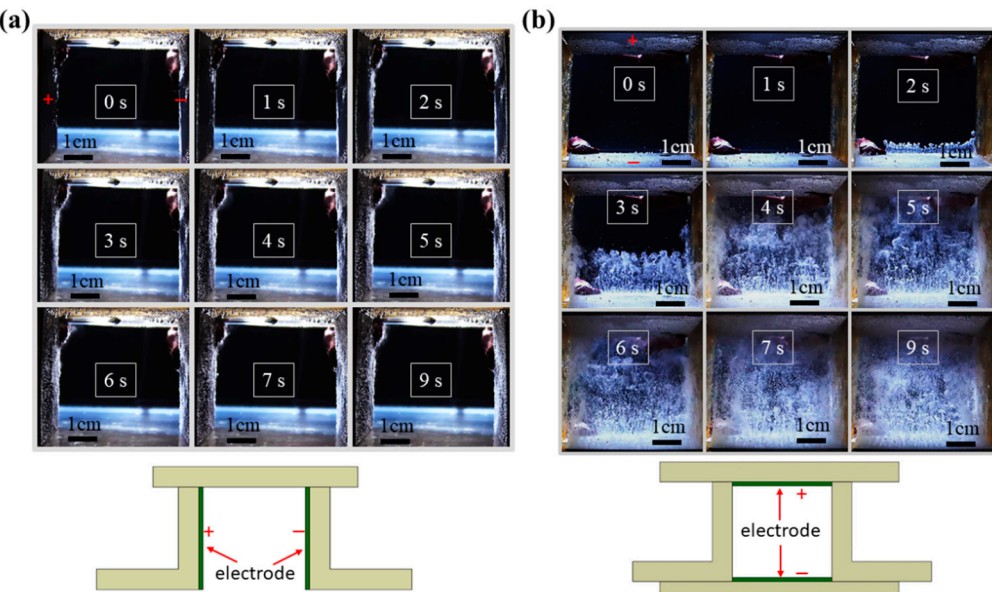

**Figure 1.** Sequential images of the evolution of gas bubbles on the surface of platinum electrodes that are placed in (**a**) a vertical and (**b**) a horizontal layout without a magnet (voltage: 200 V; electrode distance: 5 cm).

If the electrodes are placed in a horizontal layout, as shown in Figure 1b, the speed and the number of bubbles that detach from the electrode surface increase significantly. Bubbles apparently evolve from the electrode surface at nearly t = 2 s. The hydrogen bubbles swiftly detach from the cathode surface (lower electrode) and move upward due to the buoyancy force. The space between the two electrodes is almost filled with bubbles starting at the time of t = 7 s. The experimental results show that an electrode with a horizontal layout can enhance the detachment of the bubble due to the buoyancy force. However, an increase in the number of bubbles in the duct may result in a larger nonconductive space and reduce the efficiency of electrolysis.

Figure 2 shows that the current density increases almost linearly as the charging voltage is increased. It is known that platinum-based materials are excellent electrodes for hydrogen evolution reactions because they feature a higher exchange current density for the Gibbs free energy of atomic hydrogen adsorption ($\triangle G_H$) [31]. There is no doubt that a platinum electrode has a higher current density than a graphite electrode for the same charging voltage if no magnetic force is applied. At a charging voltage of 200 V, the current density increases from 20.4 mA/cm$^2$ to 26.8 mA/cm$^2$ if the vertical platinum electrode is changed to a horizontal placement. On the other hand, a graphite electrode

with a horizontal layout has a current density of 21.6 mA/cm$^2$, which is higher than the 16.8 mA/cm$^2$ for the vertical placement and the 20.4 mA/cm$^2$ for a vertical platinum electrode. This shows that an electrode that is placed horizontally allows more efficient hydrogen production.

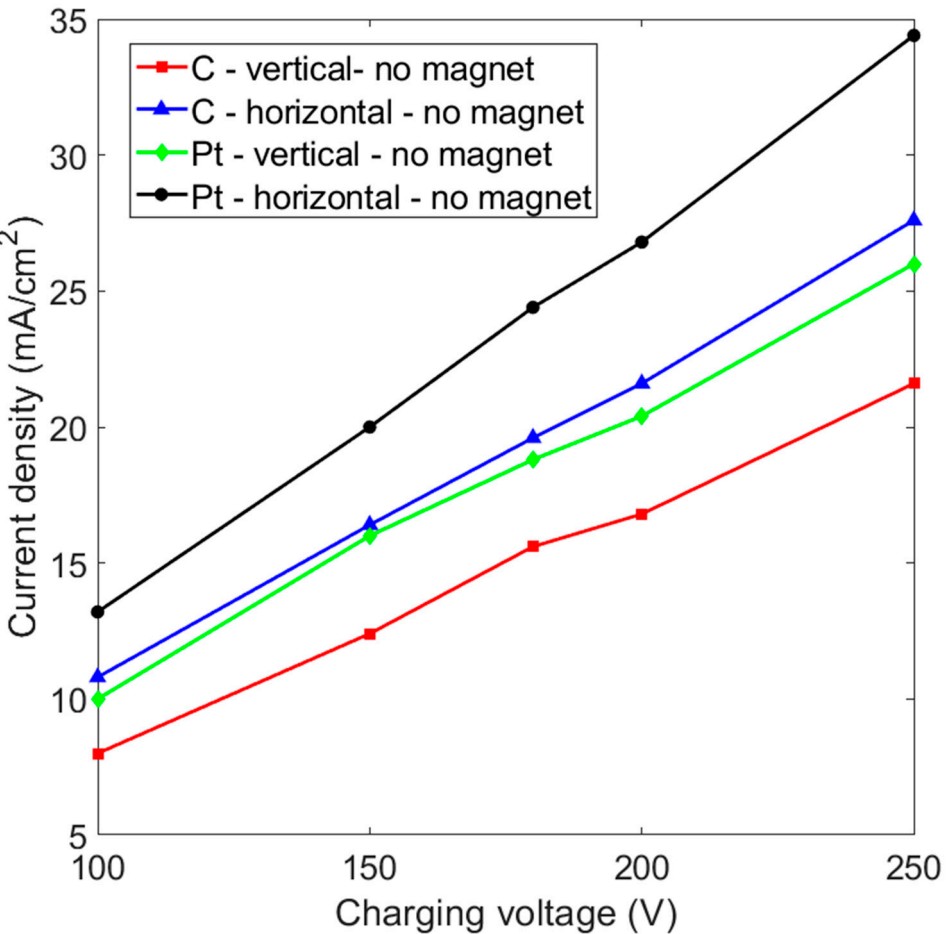

**Figure 2.** Current density vs. charging voltage for graphite (C) and platinum (Pt) electrodes that are placed in a vertical and horizontal layout without a magnet: the inter-electrode distance is 5 cm.

### 3.2. Effect of Magnetism on the Movement of Gas Bubbles

The locomotion of gas bubbles that evolve from an electrode with different magnetism is affected by the magnetohydrodynamic (MHD) effect [32–34]. MHD convection is caused by the interaction between the magnetic field and local current density, which is driven by the Lorentz force ($F_L$):

$$F_L = j \times B \tag{4}$$

where j is current density, and B is magnetic strength. Based on the configuration of the magnet and electrode in this study, the magnetic field (B) is inevitably non-uniform at the edge of the magnet and results in a gradient magnetic field at the edge of the electrode. As a result, there is normally an MHD flow around the rim when the current (j) is perpendicular to the electrode surface. The different magnetic properties of the electrode may cause various magnitudes of MHD force and flow patterns around the rim. The various locomotion behaviors of the hydrogen bubbles evolving from electrodes with different magnetism are shown in Figure 3. Figure 3a shows that hydrogen bubbles move upward from the surface of a graphite electrode when a voltage of 200 V is applied. Figure 3b shows that hydrogen bubbles from a graphite electrode move slightly towards the upper right if the S-pole of a magnet faces the cathode. This motion indicates that a

magnetic field may detach bubbles from the surface of a graphite electrode because of the magnetohydrodynamic (MHD) flow around the edge of the electrode. Figure 3c shows that hydrogen bubbles move more significantly towards the right when the graphite electrode is changed to a platinum electrode, which is a paramagnetic material with a higher magnetic susceptibility than graphite. This result shows that the paramagnetic properties enhance the locomotion of bubbles, and bubbles detach from the electrode more quickly.

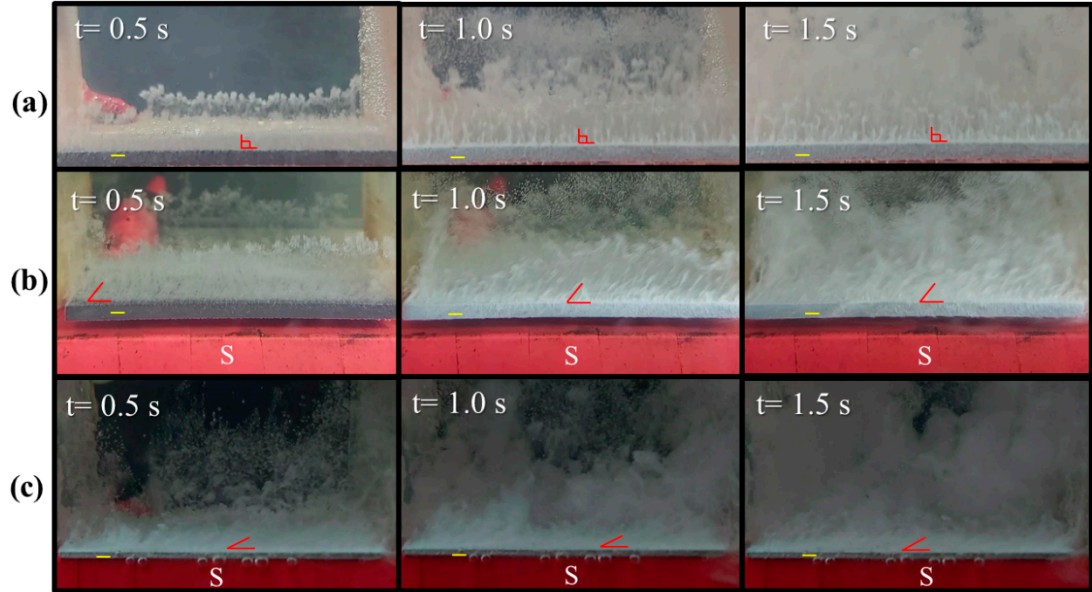

**Figure 3.** Sequential images of hydrogen bubbles that detach from (**a**) a graphite electrode without a magnet, (**b**) a graphite electrode that faces the S-pole of a magnet, and (**c**) a platinum electrode that faces the S-pole of a magnet: the inter-electrode distance is 5 cm.

Figure 4 shows the schematic diagram for the magnetic field vector between two parallel magnets. Due to the magnetic fringing effect, the direction of the magnetic field will have a horizontal component and be orthogonal to the electrical field near the edge of the electrode surface. Therefore, the Lorentz force parallel to the electrode is generated to facilitate the movement of the gas bubbles near the edge of the electrode, as shown in Figure 3b,c.

Different magnetic materials result in different degrees of magnetization. The magnetic field of the platinum electrode is stronger than that of the graphite one because of its paramagnetism. As a result, the Lorentz force produced by the platinum electrode will be more significant than that produced by the graphite electrode in the presence of a magnetic field, and the bubbles detached from the platinum electrode move faster than those from the graphite electrode.

The comparison of current density shown in Figure 5a indicates that the magnetic force has a negative effect on the efficiency of electrolysis if two horizontal electrodes are separated by more than 5 cm. The images in Figure 3 show that the magnetic field increases the detachment of the bubble from the electrode. However, bubbles are pushed upward and can stay in the duct for a longer time because there is a larger space between the electrodes. The larger number of bubbles in the duct results in higher resistance and a decrease in the efficiency of electrolysis. Figure 5b shows that the conductivity of the fluid is almost constant as the charging voltage increases if the inter-electrode distance is 5 cm. It is noted that the conductivity of graphite and platinum electrodes decreases if a magnetic field is employed. These results indicate that a magnetic field has no effect on the conductivity of a paramagnetic (platinum) or diamagnetic (graphite) material if electrodes are farther apart.

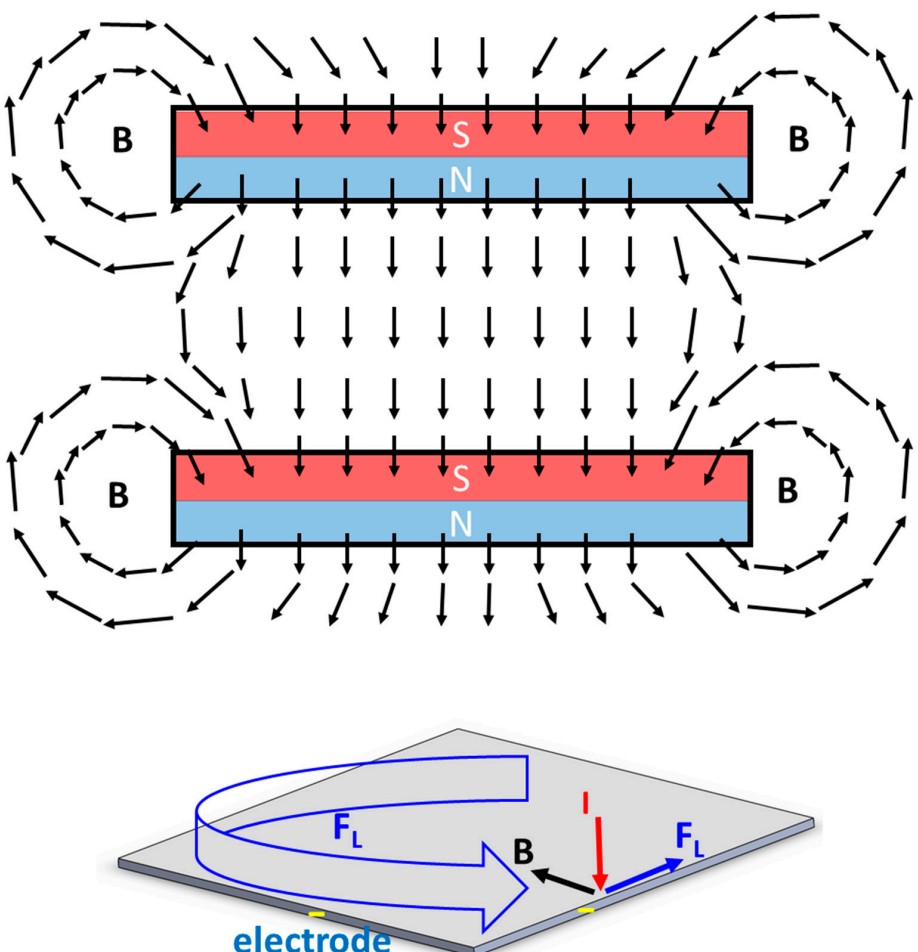

**Figure 4.** Schematic diagram of the magnetic field vector between two parallel magnets and the direction of Lorentz force near the edge of the electrode when a perpendicular electrical field is applied.

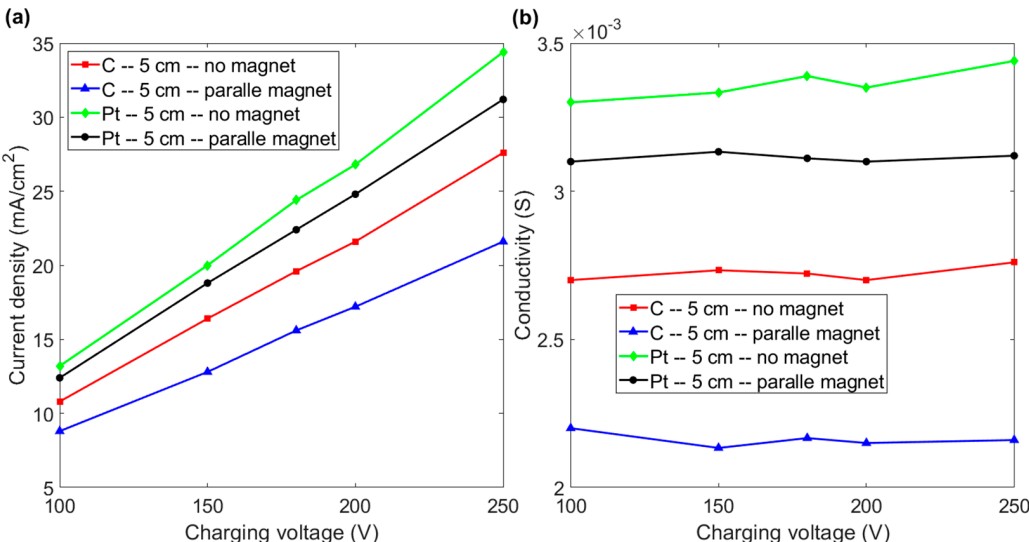

**Figure 5.** (**a**) Current density vs. charging voltage for graphite and platinum electrode with and without a magnet for an inter-electrode distance of 5 cm. (**b**) Conductivity vs. charging voltage for the various experimental layouts in Figure 5a.

### 3.3. Effect of the Distance between Electrodes

For an inter-electrode distance of 2 cm, a positive effect of the magnetic field on the current density is observed. Figure 6a,b respectively show that a platinum or graphite electrode that faces a magnet generates a greater increase in the current density and conductivity of water for the same applied voltage. As shown in Figure 6a, at a voltage of 200 V, the current density increases from 56.4 to 70.2 mA/cm$^2$ if the horizontal platinum electrode faces the magnet. If a magnetic field is applied to a graphite electrode, the current density increases from 42 to 54 mA/cm2, which is close to the value for a platinum electrode without a magnetic field. Figure 6b shows that the conductivity of the fluid increases almost linearly as the charging voltage increases for an inter-electrode distance of 2 cm. At a voltage of 200 V, the respective conductivity increases by 24.46 and 28.5% for platinum and graphite electrodes. This shows that the effect of magnetism and a magnetic field on the hydrogen evolution reaction is more significant for a shorter inter-electrode distance. This is because a shorter distance between the electrodes limits the space for the aggregation of bubbles, which then rapidly spread from the electrode surface and the duct if a magnetic field is applied. The conductivity of graphite increases by more than that of platinum, so paramagnetism does not cause stronger enhancement than diamagnetism for water electrolysis using a layout with a horizontal electrode facing a magnet for an inter-electrode distance of less than 2 cm.

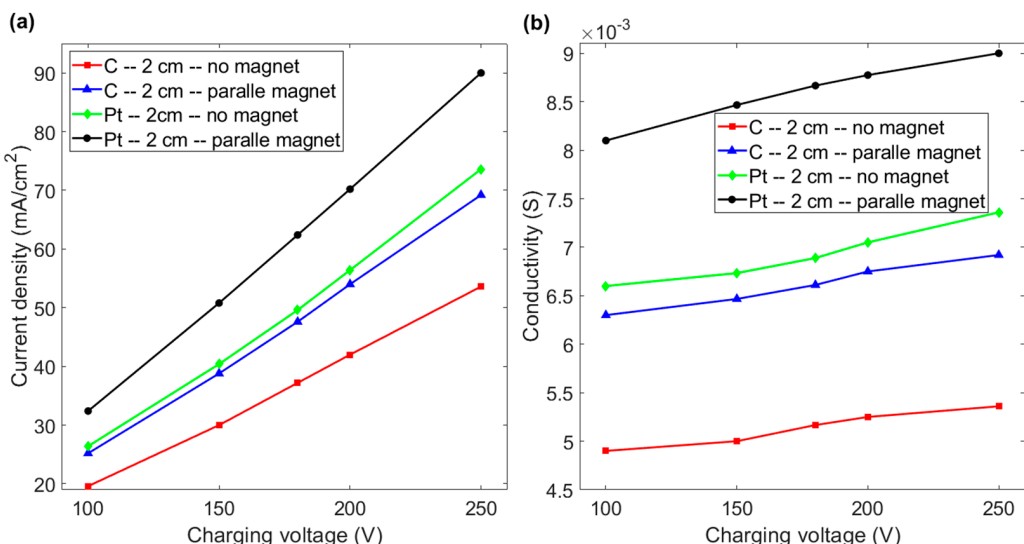

**Figure 6.** (**a**) Current density vs. charging voltage for graphite and platinum electrodes with and without a magnet for an inter-electrode distance of 2 cm. (**b**) Conductivity vs. charging voltage for the various experimental layouts in Figure 6a.

The combined effect of magnetism and inter-electrode distance is shown in Figure 7. Figure 7a shows that the current density for both graphite and platinum electrodes decreases if the distance between the electrodes is increased. For an inter-electrode distance of 5 cm, the magnetic field does not increase the conductivity of graphite or platinum electrodes, as shown in Figure 7b. For a distance of 4 cm, a paramagnetic platinum electrode exhibits increased conductivity in the presence of a magnetic field. However, subjecting graphite material to a magnetic field does not increase the conductivity until the inter-electrode distance is reduced to 2 cm. The conductivity of a graphite electrode increases by 36.4% if the inter-electrode distance is 1 cm. The conductivity of 0.0096 S is greater than the 0.008 S for a platinum electrode without a magnet, as shown in Figure 7b. These results indicate that horizontal paramagnetic and diamagnetic electrodes with a shorter inter-electrode distance in the presence of a normal magnetic field lead to a significant increase in the hydrogen production efficiency for water electrolysis.

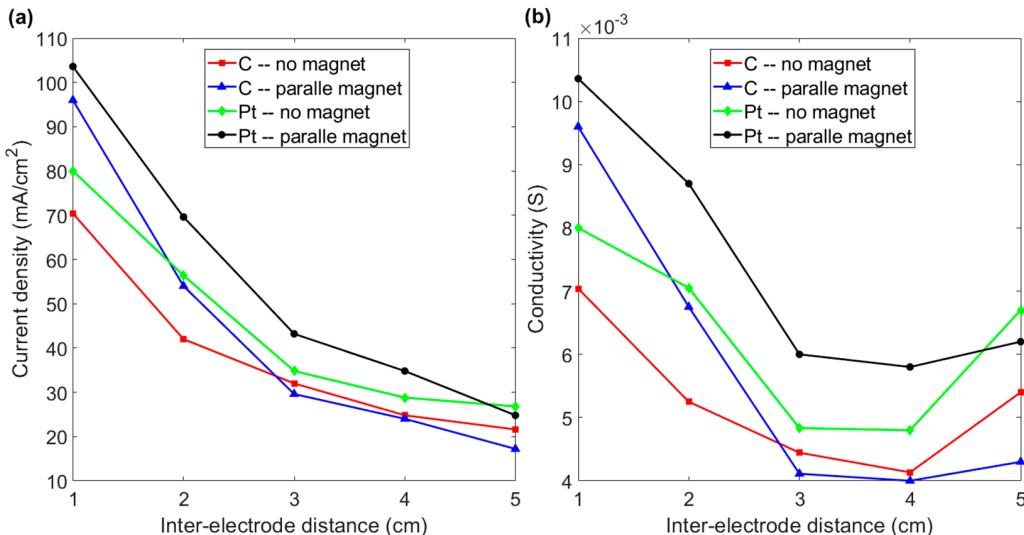

**Figure 7.** (**a**) Current density vs. inter-electrode distance for graphite and platinum electrodes with and without a magnet for various inter-electrode distances. (**b**) Conductivity vs. inter-electrode distance for the various experimental layouts in Figure 7a: the charging voltage is 200 V.

*3.4. Comparison of the Gas Production Rate for Various Experimental Layouts*

Figure 8 shows a comparison of gas production efficiency for paramagnetic and diamagnetic electrodes using inter-electrode distances of 1 to 3 cm. The production rate increases if the distance between the electrodes is reduced. If no magnetic field is applied, the vertical platinum electrode features almost the same gas production as a vertical graphite electrode for each inter-electrode distance. Table 1 shows the efficiency comparison for the accumulation of 20 mL of hydrogen for each case shown in Figures 7 and 8. The "+" sign indicates the optimal efficiency for the identical electrode material in different layouts. The "++" sign represents the best production efficiency for various materials and layouts with the same inter-electrode distance. Paramagnetic platinum electrodes that are placed horizontally and subjected to a perpendicular magnetic field produce gas most efficiently under the various experimental conditions. On the other hand, graphite electrodes with a horizontal placement that face a magnet may generate more gas than a platinum electrode without a magnet for an inter-electrode distance of less than 1 cm.

**Table 1.** Summary of the optimal hydrogen production efficiency for various electrode and magnet layouts.

| Electrode Conditions | | | Average Time (min) | Efficiency |
|---|---|---|---|---|
| **Distance (cm)** | **Material** | **Layout** | | |
| | | Vertical | 25.1 | |
| | Graphite | Horizontal | 14.25 | + |
| 5 | | Horizontal + Magnet | 16.18 | |
| | | Vertical | 23.5 | |
| | Platinum | Horizontal | 7.5 | ++ |
| | | Horizontal + Magnet | 8.35 | |
| | | Vertical | 10.6 | |
| | Graphite | Horizontal | 9.25 | + |
| 4 | | Horizontal + Magnet | 9.9 | |
| | | Vertical | 9.1 | |
| | Platinum | Horizontal | 6.7 | |
| | | Horizontal + Magnet | 5.2 | ++ |

**Table 1.** *Cont.*

| Electrode Conditions | | Average Time (min) | Efficiency |
|---|---|---|---|
| **3** | | | |
| | Graphite | Vertical — 8.37 | |
| | | Horizontal — 5.4 | + |
| | | Horizontal + Magnet — 5.7 | |
| | Platinum | Vertical — 7.9 | |
| | | Horizontal — 4.45 | |
| | | Horizontal + Magnet — 3.26 | ++ |
| **2** | | | |
| | Graphite | Vertical — 6.35 | |
| | | Horizontal — 4.62 | |
| | | Horizontal + Magnet — 3.2 | + |
| | Platinum | Vertical — 6.25 | |
| | | Horizontal — 2.51 | |
| | | Horizontal + Magnet — 2.02 | ++ |
| **1** | | | |
| | Graphite | Vertical — 3.71 | |
| | | Horizontal — 2.62 | |
| | | Horizontal + Magnet — 1.33 | + |
| | Platinum | Vertical — 3.4 | |
| | | Horizontal — 1.42 | |
| | | Horizontal + Magnet — 1.21 | ++ |

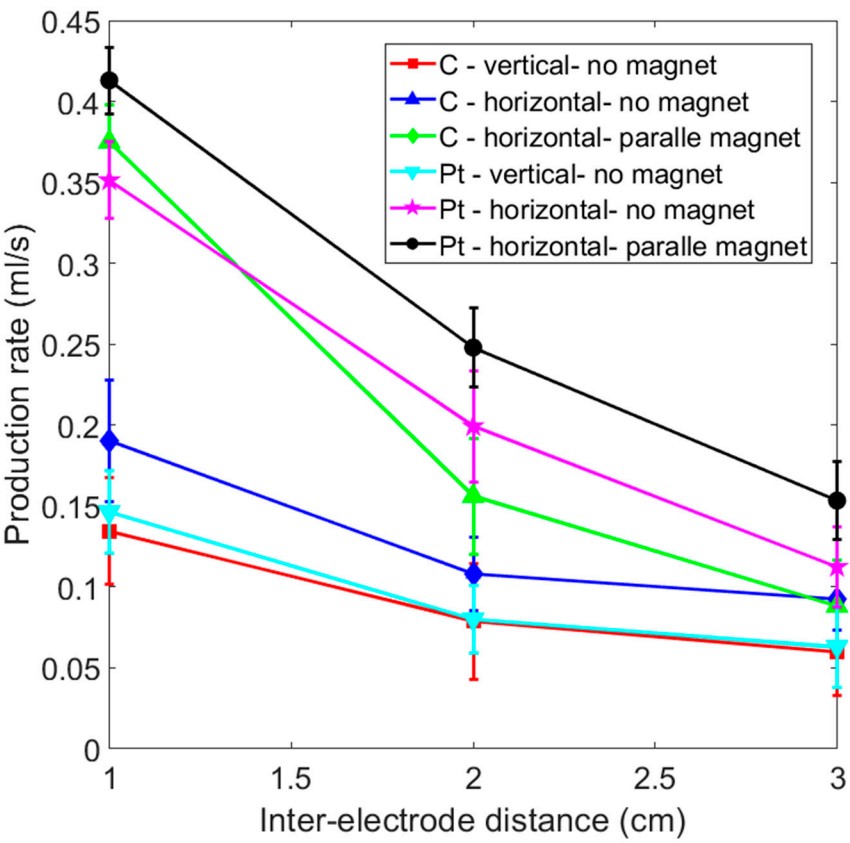

**Figure 8.** Comparison of the gas production rate for graphite and platinum electrode for various experimental layouts.

## 4. Conclusions

This study determined the effect of magnetism and a normal magnetic field on gas production efficiency. Bubbles that evolve from an electrode with a horizontal placement detach faster than bubbles from a vertical electrode. More bubbles detach from the surface of the electrode if a horizontal electrode faces a magnet. A paramagnetic (platinum) electrode has a more significant effect on bubble locomotion than a diamagnetic (graphite) electrode.

However, a magnetic field does not increase the current density for electrodes at a longer inter-electrode distance. The conductivity of a platinum electrode that faces a magnet increases when the inter-electrode distance is reduced to 4 cm, while the conductivity of a graphite electrode does not increase until the inter-electrode distance is reduced to 2 cm. Horizontal graphite electrodes that face a magnet may generate more gas than a platinum electrode without a magnet if the inter-electrode distance is less than 1 cm. These results suggest an economical way to improve hydrogen production efficiency by using a graphite electrode, which is cheaper and more abundant than platinum materials.

**Author Contributions:** Data curation, resources, methodology, writing original draft preparation, and investigation, Y.-J.C.; conceptualization, formal analysis, revising/editing the original draft, validation, and supervision, Y.-H.L.; reviewing and editing, C.-Y.C. All authors have read and agreed to the published version of the manuscript.

**Funding:** This research was funded by Ministry of Science and Technology of the Republic of China (Taiwan) with grant numbers MOST 109-2221-E-606-003 and MOST 110-2221-E-606-012.

**Institutional Review Board Statement:** Not applicable.

**Informed Consent Statement:** Not applicable.

**Data Availability Statement:** The data that support the findings of this study are available from the corresponding author upon reasonable request.

**Conflicts of Interest:** The authors declare no conflict of interest.

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
