# Peer review of "Studying the Effect of Electrode Material and Magnetic Field on Hydrogen Production Efficiency"

_magnetochemistry, doi:10.3390/magnetochemistry8050053_

Round 1

Reviewer 1 Report

This is interesting and well written paper, but certain corrections are necessary before the final decision.

Authors should give additional explanations about effect of imposed magnetic field on paramagnetic and diamagnetic materisls. It is well known that the largest effect is realized on ferromagnetic  material.  Also, Authors should pay additional attention about effect of Lorenz force. This force shows the largest effect when it is parallel to the electrode surface. Opposite, when it is a perpendicular to the electrode surface, the effect of this force is zero. The existence of some other phenomena should be adressed and discussed.

Reviewer 2 Report

The manuscript titled "Studying the effect of electrode material and magnetic field on hydrogen production efficiency" by Chen et al. reported that studying the hydrogen production efficiency by using magnetic field and results reveals that horizontal graphite electrodes that are subjected to a perpendicular magnetic field would generate a higher gas production rate. Therefore, present work is very interesting, there is a need for minor revision before publication in Magnetochemistry.

General comment

  1. In the introduction part, the authors should explore the importance of hydrogen production and the usage of applied magnetic fields. Some important green hydrogen production references are missing, such as 10.1016/j.apcatb.2021.120752; 10.3389/fenrg.2019.00015; and 10.1016/j.enchem.2021.100055
  2. Authors should provide the novelty of the present work as compared to 10.1038/s41598-021-87947-9.
  3. The author can provide details on electrode fabrication and device fabrication.
  4. Authors can provide a detailed explanation of gas bubbles detachment with an applied magnetic field for hydrogen production.
  5. The conclusion part should be more elaborate and precise to reflect the importance of the present studies towards hydrogen production.
